physiology/computational biology/medical computing

physical exercise, HIIT, young people, health

**Author for correspondence:**
Ana Cristina Silva Rebelo
e-mail: anacristina.silvarebelo@gmail.com

# Exponential model for analysis of heart rate responses and autonomic cardiac modulation during different intensities of physical exercise

Lucas Raphael Bento Silva[1,2,3],

Paulo Roberto Viana Gentil[2,3], Thomas Beltrame[6,7,8],

Marco Antônio Basso Filho[9], Fagner Medeiros Alves[2],

Maria Sebastiana Silva[2,3], Gustavo Rodrigues Pedrino[4],

Rodrigo Ramirez-Campillo[10], Victor Coswig[11]

and Ana Cristina Silva Rebelo[3,5]

[1]Department of Physical Education, Faculty Araguaia, Goiânia, Goiás, Brazil
[2]Faculty of Physical Education and Dance, [3]School of Medicine, [4]Center for Neuroscience and Cardiovascular Research, Institute of Biological Sciences, and [5]Department of Morphology, Biological Sciences Institute, Federal University of Goiás, Goiânia, Brazil
[6]Institute of Computing, University of Campinas, Campinas, Sao Paulo, Brazil
[7]Department of Physiotherapy, Federal University of Sao Carlos, Sao Carlos, Sao Paulo, Brazil
[8]Department of Physiotherapy, Universidade Ibirapuera, Sao Paulo, Sao Paulo, Brazil
[9]Department of Physiotherapy, School of Social Sciences and Health, Pontifical Catholic University of Goiás, Goiânia, Brazil
[10]Laboratory of Human Performance, Research Nucleus in Health, Physical Activity and Sport, GIAP in Quality of Life and Human Well-Being, Department of Physical Activity Science, Universidad de Los Lagos, Osorno, Chile
[11]College of Physical Education, Federal University of Pará, Castanhal, Brazil

LRBS, 0000-0002-5629-0006; PRVG, 0000-0003-2459-4977;
TB, 0000-0001-6568-0869; MSS, 0000-0001-7265-5872;
GRP, 0000-0003-0488-5400; RR-C, 0000-0003-2035-3279;
VC, 0000-0001-5461-7119; ACSR, 0000-0002-9214-5025

The aim of this study was to compare the heart rate (HR) dynamics and variability before and after high-intensity interval training (HIIT) and moderate-intensity continuous training (MICT) protocols with workloads based on treadmill workload at which maximal oxygen uptake was achieved (WL$\dot{V}O_2$max). Ten participants performed cardiopulmonary

exercise testing (CPET) to obtain oxygen uptake (WL$\dot{V}O_2$max). All training protocols were performed on a treadmill, with 0% grade, and had similar total distance. The MICT was composed by 21 min at 70% of WL$\dot{V}O_2$max. The first HIIT protocol (HIIT-30 : 30) was composed by 29 repetitions of 30 s at 100% of s$\dot{V}O_2$max and the second HIIT protocol (HIIT-4 : 3) was composed by three repetitions of 4 min at 90% of WL$\dot{V}O_2$max. Before, during and after each training protocol, HR dynamics and variability (HRV) were analysed by standard kinetics and linear (time and frequency domains). The repeated measures analysis of variance indicated that the HR dynamics, which characterizes the speed of HR during the rest to exercise transition, was statistically ($p < 0.05$) slower during MICT in comparison to both HIIT protocols. The HRV analysis, which characterizes the cardiac autonomic modulation during the exercise recovery, was statistically higher in HIIT-4 : 3 in comparison to MICT and HIIT-30 : 30 protocols ($p < 0.005$ and $p = 0.012$, respectively), suggesting that the HIIT-4 : 3 induced higher sympathetic and lower parasympathetic modulation during exercise in comparison to the other training protocols. In conclusion, HIIT-4 : 3 demonstrated post-exercise sympathetic hyperactivity and a higher HRpeak, while the HIIT-30 : 30 and MICT resulted in better HRV and HR in the exercise-recovery transition. The cardiac autonomic balance increased in HIIT-30 : 30 while HIIT-4 : 3 induced sympathetic hyperactivity and cardiac overload.

# 1. Introduction

There is a wide debate regarding optimal exercise training protocols to improve cardiorespiratory function, glycaemic control, disease prevention and treatment [1] as well as to reduce mortality [2]. The World Health Organization (WHO) recommends 150 min week$^{-1}$ of moderate intensity or 75 min week$^{-1}$ of vigorous physical activities [3]. Although it is a common practice to prescribe moderate-intensity continuous training (MICT) to meet the WHO standards, there is evidence suggesting that high-intensity interval training (HIIT) promotes equivalent or even better physiological benefits than those achieved with MICT [4–6]. However, considering the wide range of possible parameter combinations for HIIT protocol designs, this training modality is also being debated [7–9]. Therefore, it is necessary to compare different HIIT protocols with MICT to provide a better physiological understanding of the effects of such training protocols and allow a better exercise training prescription [10].

Among the possible effects and outcomes of an exercise intervention, the study of the cardiac autonomic nervous system (ANS), by the analysis of the heart rate (HR) responses during the transitions from rest to exercise to recovery, provides information that might be used in the future for a better training protocol prescription, avoiding the risks and controlling the onset of fatigue [11–14]. During the rest to exercise transition, the HR increases initially due to a reduction of the parasympathetic nervous system activation, followed by an increase in the sympathetic tone [15]. During the recovery, the HR initially decreases based on a sudden vagal reactivity followed by a gradual sympathetic autonomic tone reduction [16]. The evaluation of the ANS by characterizing HR responses during the rest, exercise and recovery periods presents an interesting prognostic value in the general population [17–20] and can be used as a cardiovascular fitness index [15,21].

One popular HIIT protocol is composed of 30 s of exercises at 100% of the workload reached during the maximal oxygen uptake (WL$\dot{V}O_2$max), with resting intervals of 30 s between bouts (HIIT-30 : 30) [22]. Another popular HIIT protocol is composed of 4 min of exercises between 90% and 95% of the peak heart rate (HRpeak) with recovery intervals of 3 min at 50–70% (HIIT-4 : 3) of the HRpeak [23]. A previous study showed that the former results in higher oxygen consumption, higher HR and higher rate of perceived exertion [24]. Confirming this, previous studies showed that HIIT with longer duration seems to be more stressful to the cardiovascular system. However, its effects in HR dynamics and variability are not known [25,26].

This study aimed to compare HR responses as a surrogate of ANS balance during three different exercise training protocols (two HIIT and one MICT) in healthy young male individuals. We hypothesized that HIIT protocols are capable of promoting better HR adjustments in healthy young men and that such protocols can be safely performed, promoting benefits like other intensities.

# 2. Methods

## 2.1. Participants

This was a randomized crossover study. Nineteen volunteers initiated the study. Inclusion criteria were: (i) being between 18 and 30 years old; (ii) body mass index (BMI) between 18.5 and 29.9 kg m$^{-2}$, (iii) being

physically active for at least six months before the study, and (iv) not consuming any type of stimulants during the execution of the protocols. And the exclusion criteria were: (i) presenting uncontrolled arrhythmia during physical exertion or other type of cardiovascular disease, (ii) chronic obstructive pulmonary disease, (iii) cancer and/or kidney disease, (iv) neurological deficit, and (v) osteomioarticular limitations.

Seven participants were excluded ($n = 7$) due to difficulty in performing the exercise protocols and two were excluded because they presented irregular data for analysis. Therefore, the final sample consisted of 10 males (age of $26.9 \pm 3.95$ years; BMI of $24.1 \pm 1.99$ kg m$^{-2}$). This study was carried out in accordance with the recommendations of the University Ethics Committee, under number 1.643.562, and all subjects gave written informed consent in accordance with the Declaration of Helsinki.

All participants visited the laboratory on four different occasions interspersed by 48–72 h. The first day involved recording the medical history, anthropometric measurements (body mass and height) and clinical data (blood pressure by the Korotkoff auscultatory method repeated every 2 min during the test period) and heart rate during resting. During the same visit, cardiopulmonary exercise testing (CPET) was performed to obtain the maximal oxygen uptake ($\dot{V}O_2max$) and the workload of $\dot{V}O_2max$ (WL$\dot{V}O_2max$). During the subsequent three visits, participants performed either HIIT-4 : 3, HIIT-30 : 30 or MICT in a randomized order. The randomization process occurred online, through the website: https://www.random.org/.

## 2.2. Cardiopulmonary exercise testing

The CPET and all physical exercise protocols were performed on the same electric treadmill (Centurion 200, Micromed, Brazil) connected to a personal computer. Participants remained at rest in the orthostatic position for 3 min before the CPET, followed by a 2 min warm-up at 5 km h$^{-1}$. Afterwards, the treadmill speed was increased by 1 km h$^{-1}$ every minute until volitional exhaustion. All tests were performed without inclination to minimize the errors during the execution of the protocol and to avoid different physiological responses. After the CPET, an active recovery period was performed for 2 min at 2 km h$^{-1}$, followed by 4 min of recovery in the sitting position.

HR was monitored using a cardiac monitor (Polar V800, Finland) [22]. Where appropriate, RR intervals, in milliseconds, were calculated as $RRi = 60\,000/HR$ and HR was obtained by calculation: $HR = 60\,000/RRi$ and expressed in bpm. Ratings of perceived exertion (RPE) of the legs and of dyspnoea [25] were assessed using the Borg scale at the end of each stage of the CPET protocol [27]. Based on the breath-by-breath system model, MetaLyzer® continuously assesses volume and jointly determines the expired concentration of $CO_2$ and $O_2$. $CO_2$ production and $O_2$ absorption during each breath are calculated and the data obtained are transferred breath by breath to a laptop for real-time display (Cortex, Metalyzer II, Rome, Italy).

## 2.3. Exercise protocols

All protocols were equated for total distance covered and were executed without inclination. For each training protocol, the exercise intensities (i.e. percentage of WL$\dot{V}O_2max$) were adapted from the previous studies, as described in table 1 [23,28]. For all training protocols, volunteers performed a 5 min warm-up at 55% of WL$\dot{V}O_2max$. At the end of each training session, volunteers performed a 3 min of cool-down at 50% of WL$\dot{V}O_2max$ (table 1).

## 2.4. Heart rate variability (HRV) recording and analysis

All volunteers were evaluated in the morning to avoid differences in physiological responses due to circadian rhythm. The measurements were made in an air-conditioned room, with temperatures ranging from 22°C at 24°C and relative humidity between 40 and 60%. The volunteers were instructed to not consume caffeine or alcohol nor perform any type of physical exercise in the 24 h preceding the test. The volunteers were also instructed to avoid copious meals and to have a light meal at least two hours before the test. RR intervals were recorded at rest and during exercise. At rest the participants were in dorsal decubitus and breathing normally, recording occurred over a period of 8 min using a HR monitor (Polar® V800, Polar Electro Oy, Kempele, Finland). During the physical exercise session, the RR intervals were collected before the beginning of the physical exercise for 3 min in the pre-exercise position, throughout the protocol and after the interruption for 4 min also in the pre-exercise position.

**Table 1.** Protocols applied to study participants. WL$\dot{V}O_2$max = Workload reached at the time of reaching the maximum oxygen consumption ($\dot{V}O_2$max).

| protocol | warm-up | exercise | recovery | cool-down | total time | total distance |
|---|---|---|---|---|---|---|
| HIIT-4 : 3 | 5 min—55% of WL$\dot{V}O_2$max | 3 × 4 min—90% of WL$\dot{V}O_2$max | 2 × 3 min—60% of WL$\dot{V}O_2$max | 3 min—50% of WL$\dot{V}O_2$max | 26 min of exercise | 3600 m |
| HIIT-30 : 30 | 5 min—55% of WL$\dot{V}O_2$max | 29 × 30 s—100% of WL$\dot{V}O_2$max | 30 s passive | 3 min—50% of WL$\dot{V}O_2$max | 37 min of exercise | 3625 m |
| MICT | 5 min—55% of WL$\dot{V}O_2$max | 21 min continuous— 70% of WL$\dot{V}O_2$max | | 3 min—50% of WL$\dot{V}O_2$max | 32 min of exercise | 3675 m |

During this period, the HRV was analysed by linear regression (time and frequency domains). The HR data portion, with 256 consecutive beats, with the greatest stability of the RRi time series was selected for the HRV analysis [29].

The time domain parameters studied were the standard deviation of all RRi (SDNN, presented in equation (2.1)) and the square root of the mean squared differences between adjacent RRi (RMSSD, presented in equation (2.2)). The SDNN reflects overall HRV, while the RMSSD can be considered an index of cardiac parasympathetic modulation.

$$\text{SDNN} = \sqrt{\frac{1}{N-1}\sum_{j=1}^{N}(\text{RR}_j - \overline{\text{RR}})^2} \tag{2.1}$$

and

$$\text{RMSSD} = \sqrt{\frac{1}{N-1}\sum_{j=1}^{N-1}(\text{RR}_{j+1} - \text{RR}_j)^2}, \tag{2.2}$$

where $\text{RR}_j$ denotes the value of the $j$-th interval RR, $N$ is the total number of successive intervals and $\overline{\text{RR}}$ represents the mean of the RR intervals.

Spectral analysis was performed using fast Fourier transformation [14]. Responses at higher frequencies (HFs), from 0.15 to 0.40 Hz, reflect the activity of the vagus nerve (as an index of parasympathetic activation). On the other hand, responses at lower frequencies (LFs), from 0.04 to 0.15 Hz, reflect sympathetic activity [30]. The LF/HF ratio was also calculated to determine the sympathovagal balance [29,31].

After synchronization, the R-R intervals were examined visually for artefacts. The artefacts identified during physical exercise sessions were classified into two types, the first: it happens when a beat is approximately 30% greater than the value of the previous R-Ri. The second type: happens when the noise is also seen as a beat and causes an interval 30% smaller than the previous one [32].

## 2.5. Exponential data modelling

The HR dynamics during the entire rest-to-exercise transition and during the recovery period were analysed during the CPET, HIIT and MICT protocols. The HR data at the off-transient were collected at the end of CPET and then filtered and analysed with an ad hoc routine developed using OriginPro 8.0 software (OriginLab, Northampton, MA, USA). This algorithm applies an exponential model to the data corresponding to the full recovery period [33]. A nonlinear algorithm that minimizes the sum of squared errors as a convergence criterion was used to determine the best parameters for the resulting exponential curve [34]. The function was only included in the final analysis if $r > 0.95$. The kinetics were modulated using the following time exponential function (equations (2.3) and (2.4)) [35]:

$$\text{Transition on: } Y(t) = Y_{(\text{LB})} + A(1 - \text{e}^{-(t-TA)/\tau}) \tag{2.3}$$

and

$$\text{Transition off: } Y(t) = Y_{(\text{LB})}(A.\text{e}^{-(t-TA)/\tau}) + Y_{(\text{LB})}, \tag{2.4}$$

where '*t*' is time, 'HRpeak' is the peak HR at the end of CPET, '*A*' is the amplitude of HR reduction after the end of exercise, '*τ*' is the exponential time constant and 'TA' is a time delay. The inclusion of 'TA' was due to the possibility of HRoff not decreasing immediately after load interruption.

The kinetics analysis parameters (i.e. the speed of the adaptation, or *τ*, and the steady-state amplitude, or *A*) were calculated using a computer program created in LabVIEW (National Instruments, Austin, TX) following standard procedures [36]. The fitting quality was assessed by the analysis of residuals, correlation coefficients and the 95% confidence interval band [37,38].

## 2.6. Delta analysis

The HRR data were also characterized by HR deltas analysis. The selected intervals of the recovery period (0–180 s, every 30 s) were subtracted from HRpeak, presented in equation (2.4). The intervals were measured by calculating the mean HR in the 5 s before and after each time point. The higher the delta was, the faster the HRR adjustment because HR fell more after a fixed period [39].

$$\Delta_{(x)} = \tau - \text{HRpeak}. \tag{2.5}$$

## 2.7. Statistical analysis

The sample size calculation was estimated from the GraphPad StatMate for Windows 2.0 software (GraphPad Software, La Jolla, CA, USA) based on the mean and standard deviation of the RMSSD and delta 30 variables obtained in a pilot study and the software was also used for generating high quality/resolution graphics. For an alpha of 0.05 and a power of 80%, the recommendation was 10 volunteers. Values of $p < 0.05$ were considered significant. The normal distribution of the data was evaluated by means of the Kolmogorov–Smirnov test. When the normality of the samples was verified, the repeated measures of analysis of variance (ANOVA) were used, and when non-normal distribution was found the Kruskal–Wallis test was used to compare the differences between the variables obtained in each protocol of exercise. Statistical analysis was performed in the statistical program Statistical Package for the Social Sciences (SPSS; IBM Corp., Armonk, NY, USA), version 21.

## 3. Results

Table 2 presents the characteristics data of the volunteers.

HRpost presented lower values in the HIIT-30 : 30 and MICT protocols than in the HIIT-4 : 3 protocol ($p = 0.015$ and $p = 0.016$, respectively) (table 3). The HIIT-4 : 3 protocol presented higher values of HRpeak than the HIIT-30 : 30 and MICT ($p < 0.0001$ and $p = 0.005$, respectively) protocols. There was no significant difference between HIIT-30 : 30 and MICT.

When analysing the HRon kinetic variables, a significant difference was found in the *τ* variable between the MICT and HIIT-4 : 3 protocols ($p = 0.041$), with MICT promoting a delay in the HR response in the transition between rest and exercise. The same result was obtained in the comparison between the MICT and HIIT-30 : 30 protocols ($p = 0.032$), since MICT presented a longer HR response time than the HIIT-30 : 30 protocol (table 3). There were no differences between HRon for HIIT-30 : 30 and HIIT-4 : 3. In the comparison of the differences between HR deltas, the HIIT-4 : 3 protocol had the lowest HR reduction in the initial 60 s of recovery when compared to that in the HIIT-30 : 30 and MICT protocols, represented by Δ30 ($p < 0.001$ and $p = 0.034$, respectively) and Δ60 ($p = 0.012$ and $p = 0.037$, respectively). In the comparison between HIIT-30 : 30 and MICT, there was only a difference in Δ30 ($p = 0.032$), evidencing a greater decrease in the HIIT-30 : 30.

Table 4 presents the results of HRoff analysis immediately after the end of the exercise. Only the LF and HF variables were different between HIIT-4 : 3 and MICT ($p < 0.005$ and p = 0.012, respectively), with the MICT protocol showing higher LF and lower HF post-exercise. There was no difference among the protocols in any other parameter.

Figure 1 shows the data of the monoexponential HRon and HRoff analysis. In the HRon, the protocol MICT presented higher values of the *τ* variable (called a 'time constant', used to describe the progression or decline of heart rate), which means a greater slowness of HRon kinetics than in the HIIT-30 : 30 and HIIT-4 : 3 protocols ($p < 0.05$). When comparing HRoff data between the three protocols, HIIT-30 : 30 presented lower values of the *τ* variable than HIIT-4 : 3 and MICT ($p < 0.05$).

**Table 2.** HR variability and kinetics in participants. Data are presented as mean $\pm$ standard deviation; SDNN = standard deviation of all normal RR intervals recorded in a time interval; rMSSD = square root of mean squared differences between adjacent normal RR intervals, in a time interval; pNN50 = proportion of NN50 divided by the total number of NNs; HF = high frequency; LF = low frequency; HF/LF = ratio; un = normalized units; Amp = amplitude; $\tau$ = time constant.

| | $(n = 10)$ |
|---|---|
| autonomic heart modulation | |
| rest | |
| SDNN (ms) | $42.5 \pm 18.2$ |
| rMSSD (ms) | $43.6 \pm 16.1$ |
| pNN50 | $23.5 \pm 18.8$ |
| HF (un) | $43.8 \pm 22.1$ |
| LF (un) | $56.0 \pm 22.2$ |
| HF/LF | $1.99 \pm 1.5$ |
| | |
| HR kinetics | |
| during exercise | |
| HRon | |
| Amp | $46.7 \pm 18.7$ |
| $\tau$ | $35.9 \pm 16.7$ |
| | |
| HRoff | |
| Amp | $114.3 \pm 20.6$ |
| $\tau$ | $101.6 \pm 25.1$ |

**Table 3.** Cardiovascular variables and linear HR analysis in each protocol. Data are presented as mean $\pm$ standard deviation. HR = heart rate; SBP = systolic blood pressure; DBP = diastolic blood pressure.

| $(n = 10)$ | HIIT-4 : 3 | HIIT-30 : 30 | MICT |
|---|---|---|---|
| cardiovascular variables | | | |
| HR before, bpm | $61.0 \pm 9.59$ | $63.4 \pm 8.73$ | $61.0 \pm 11.37$ |
| HR post, bpm | $104.2 \pm 11.81$[a,b] | $97.6 \pm 12.77$ | $96.3 \pm 14.07$ |
| HR peak, bpm | $187.2 \pm 6.23$[a,b] | $156.5 \pm 4.55$ | $171.4 \pm 5.73$ |
| SBP before, mmHg | $117.3 \pm 7.66$ | $122.6 \pm 5.01$ | $122.9 \pm 14.60$ |
| SBP post, mmHg | $127.7 \pm 10.06$[a] | $116.5 \pm 7.73$ | $124.3 \pm 7.97$ |
| DBP before, mmHg | $73.2 \pm 9.96$ | $72.6 \pm 7.76$ | $73.0 \pm 7.48$ |
| DBP post, mmHg | $76.7 \pm 9.59$ | $77.8 \pm 5.65$ | $77.3 \pm 11.19$ |
| linear analysis of HRoff | | | |
| $\Delta_{30}$ | $-5.9 \pm 5.60$[a,b] | $-15.6 \pm 10.5$ | $-8.8 \pm 6.79$ |
| $\Delta_{60}$ | $-10.0 \pm 8.32$[a] | $-18.4 \pm 8.08$ | $-13.7 \pm 7.07$ |
| $\Delta_{90}$ | $-16.8 \pm 10.97$ | $-19.9 \pm 8.53$ | $-15.9 \pm 9.07$ |
| $\Delta_{120}$ | $-26.4 \pm 10.97$ | $-22.3 \pm 5.49$ | $-18.3 \pm 8.92$ |
| $\Delta_{150}$ | $-32.4 \pm 9.32$ | $-26.6 \pm 8.57$ | $-19.7 \pm 8.59$ |
| $\Delta_{180}$ | $-37.6 \pm 8.15$ | $-25.7 \pm 23.21$ | $-22.7 \pm 11.36$ |

[a]Statistically significant—comparison between protocols 30 : 30 and 4 : 3.
[b]Statistically significant—comparison between continuous protocol and 4 : 3.
[c]Statistically significant—comparison between continuous protocol and 30 : 30.

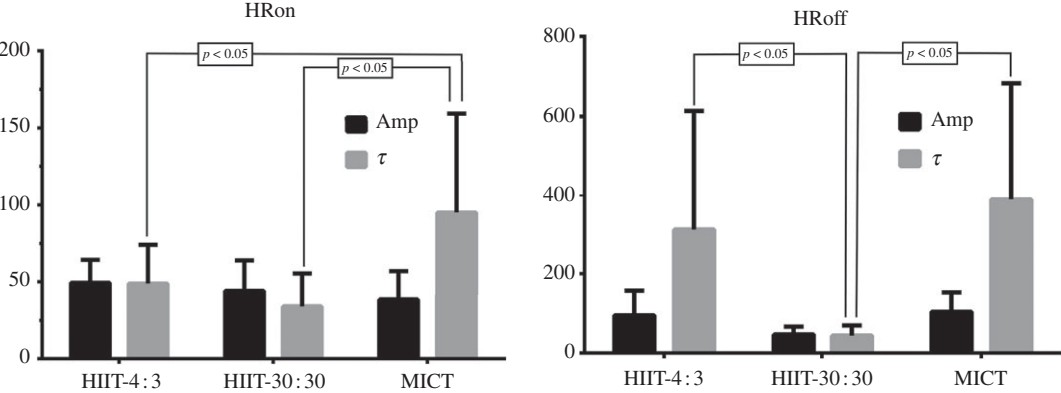

**Figure 1.** Monoexponential analysis of HRon and HRoff. Amp, amplitude; $\tau$, time constant.

**Table 4.** Analysis of post-exercise heart rate variability in the three protocols. Data presented in mean ± standard deviation. SDNN = standard deviation of all normal RR intervals recorded in a time interval, expressed in milliseconds (ms); rMSSD = square root of the square mean of the differences between the adjacent normal RR intervals, in a time interval, expressed in ms; HF = high frequency (un); LF = low frequency (un).

| (n = 10) | HIIT-4 : 3 | HIIT-30 : 30 | MICT |
|---|---|---|---|
| SDNN, ms | 11.04 ± 12.05 | 11.70 ± 12.57 | 9.16 ± 6.21 |
| rMSSD, ms | 7.46 ± 5.79 | 9.36 ± 8.14 | 9.24 ± 10.81 |
| LF, un | 81.19 ± 13.56[a,b] | 75.54 ± 17.59 | 75.01 ± 20.88 |
| HF, un | 18.87 ± 13.24[a,b] | 24.27 ± 17.33 | 24.73 ± 20.46 |

[a]Statistically significant—comparison between protocols 30 : 30 and 4 : 3.
[b]Statistically significant—comparison between continuous protocol and 4 : 3.
[c]Statistically significant—comparison between continuous protocol and 30 : 30.

## 4. Discussion

Although the term HIIT indicates 'high-intensity interval training' in a generic sense, the literature points out different possibilities for applying this type of training [8]. Several studies have investigated post-exercise HRV; however, few have compared HRV responses in different training modalities [12,40,41]. In addition to the methodological challenge there is a complexity in standardizing exercise intensity when comparing different modalities. To the best of our knowledge, this study is the first to evaluate and compare two modalities of HIIT with MICT in HR dynamics in healthy individuals. The main findings include a higher increase in HR (HRpeak) during HIIT-4 : 3. Although both HIITs had a faster HR response, HIIT-30 : 30 promoted faster HR and HRV recovery. Perhaps the greatest challenge generated by the present study is the need to classify the intensity within a specific criterion, bearing in mind that HIIT-30 : 30 was performed at lower cardiovascular intensities (HR) though at a higher mechanical intensity (speed). In contrast, HIIT-4 : 3 would involve the opposite, with higher cardiovascular intensity (HR) and lower mechanical intensity (speed).

Regarding HRon, a previous study reported that increased HR in the rest-exercise transition would be associated with increased central blood volume, promoted by the muscle contractions during exercise; this would result in a greater volume of ejection that, when detected by arterial and/or carotid baroreceptors, would send a signal to the central nervous system that would trigger a parasympathetic withdrawal and a sympathetic activation [16]. In steady-loading exercises, it is possible to detect an increase in heart rate as an extension of the duration of the execution [42].

In our study, we found that the MICT protocol promoted a slower HR response in the rest-exercise transition than that observed with the HIIT protocols, as determined by the time constant ($\tau$). This may be due to the constant workload performed at a low intensity, which would generate reduced effective participation of the sympathoadrenal system. At HIIT-30 : 30, despite the initial mechanical intensity being higher (100% of WL$\dot{V}O_2$max), the short duration of the effort seems to have generated less of a need for sympathetic activation, which may explain the lack of difference for HIIT-4 : 3.

Similar to previous findings [24], in the present study, the highest HR was found during HIIT-4 : 3 compared to the other protocols. Although the HRpeak was lower in the HIIT-30 : 30 protocol than in the MICT protocol, the differences did not reach significance.

Based on these findings, it may be suggested that longer-lasting (greater than 2 min) HIIT protocols performed in close proximity to WL$\dot{V}O_2$max promote higher cardiac stress than HIIT protocols at an equivalent or even higher mechanical intensity but with shorter duration and MICT, even when equated by workload (distance, caloric expenditure, etc.). On the other hand, the HIIT protocols with shorter durations present values equal or inferior to MICT. Thus, using short duration submaximal stimuli may be an effective strategy for decreasing cardiovascular overload in an exercise programme while working at relatively high mechanical intensities.

As for the HRoff, the delta analysis showed a greater reduction in HR in the HIIT-30 : 30 and MICT protocols after 30 s and 1 min than in the HIIT-4 : 3 protocol. Although there was a trend for this reduction to be faster after 30 : 30 than with MICT, the differences were not significant. HRoff results from the interaction of the parasympathetic reactivation and sympathetic withdrawal, with the reactivation of the parasympathetic occurring more rapidly and, therefore, playing the most important role in the early deceleration of the HR [43]. Therefore, our results suggest that HIIT-4 : 3 induces a delayed parasympathetic response.

A previous study compared the HRoff response after 16 min of HIIT protocols with stimuli of 4, 2 and 1 min at greater than 80% of $\dot{V}O_2$ peak and active rest of 2 min at 60% of $\dot{V}O_2$ peak, and MICT sessions consisting of 45 min of cycling exercise between 60 and 70% of $\dot{V}O_2$ peak. The results of this study showed a trend for lower values of the deltas of the HRoff in the continuous protocol than in the HIIT protocols performed on bike [42]. These results are partially in line with the present study, since we observed from the monoexponential analysis that the HIIT-30 : 30 protocol presented faster HRoff kinetics, as observed by the smaller values of the $\tau$ variable, than the HIIT-4 : 3 and MICT protocols ($p < 0.05$).

HRoff is an important marker of the decrease in vagal activity in healthy individuals, decreasing from approximately 30 beats in the first minute and 52 beats in the second minute after the peak of the exercise [44]. Interestingly, in our findings, only HIIT-30 : 30 showed values above 30 bpm after the first minute of recovery.

A previous study also reported that HIIT presented faster HR recovery than MICT [45]. HIIT was carried out with 1 min of stimulus and 1 min of rest; the continuous exercise was performed over 60 min [45]. Both protocols had an intensity of 70% of $\dot{V}O_2$max and were performed in a cycle ergometer. Similar results with active adults revealed slower HR recovery after low-volume sprint interval training (30 s of 'all-out' cycling with a 4 min recovery) compared to high-volume endurance training (90 to 120 min continuous cycling to approximately 65% $\dot{V}O_2$ peak) [46]. In our study, only 4 : 3 showed a slower HRR, which suggests that HRoff dynamics during HIIT depends on the protocols used.

In our study, it was possible to compare the three distinct protocols and it can be suggested that HIIT with longer stimuli results in a longer HR recovery, which suggests that HIIT with short stimuli and MICT might promote cardiac protection [7,47]. The smaller reduction of the HRoff in the 4 : 3 protocol may be due to a parasympathetic control deficit and represents an unfavourable prognosis in terms of cardiovascular risk [48,49]. The HR behaviour during exercise is sympatho-dependent in the late periods; thus, the greater the intensity of the exercise, the greater is the action of the circulating catecholamines and the afferent metaboreflex action initiated in the active skeletal muscles [50]. In addition, factors such as mechanical distension of the atrium as a function of venous return, body temperature and blood acidity influence the HR response [51].

Our results demonstrated that HIIT-4 : 3 resulted in higher sympathetic modulation (HF) and lower parasympathetic modulation (LF) than HIIT-30 : 30 and MICT, compatible with higher cardiovascular overload. These data corroborate previous studies that demonstrated that a greater intensity of exercise, when evaluated by cardiorespiratory parameters, is associated with slower recovery of the parasympathetic, specifically from the RMSSD or HF [52,53].

Our findings are in line with those of others who showed that the higher the intensity of the exercise was, measured by the HRpeak, the slower the recovery of vagal activity [54,55]. These results are in agreement with the present study in relation to HRoff data representing the vagal activity (RMSSD and HF) and show a lower reduction of the vagal component after physical exercise in HIIT-4 : 3 than in the other protocols analysed. With the increase in the cardiorespiratory intensity of the exercise, there is also an expansion in the demand of the mechanical functions of the organism associated with the amount of non-oxidizing energy expenditure and with a greater stimulation of the metaboreceptors [56–58]. Our findings suggest that there is also a predominance of sympathetic

stimulation, i.e. the sympathetic stimulus continues to act in the recovery phase, despite vagal reactivation.

Some limitations of the present study should be acknowledged. We studied healthy subjects and, considering the autonomic modulations presented by people with cardiovascular disease, their adaptations before, during and after the exercise can be different compared to healthy subjects, so the results of the present study may not be applicable in subjects with pathological conditions. Further studies should be carried out on larger, healthy and diseased populations to determine the specific indications of training prescription according to the specificities of each population. Another important point is that the study was performed on an electric treadmill and the responses might differ in other forms of exercise (bike, track running, calisthenics…). Moreover, it is important to consider that the present design allows us only to bring observations that generate hypotheses for further studies on long-term training programmes.

## 5. Conclusion

It was concluded that in the rest–exercise transition, the MICT protocol promoted a faster HR response in relation to both HIIT protocols. In contrast, HIIT-4 : 3 demonstrated post-exercise sympathetic hyperactivity and a higher HRpeak, while HIIT-30 : 30 led to faster HR recovery and higher HRV recovery, which makes its prescription seemingly safe in cardiovascular terms. As for relevance and applicability these results suggest that HIIT-4 : 3 protocol should be used to improve the cardiorespiratory performance of athletes with no history of cardiopathies, while HIIT-30 : 30 can be used to minimize cardiovascular stress and provide safer responses in terms of autonomic modulation.

Ethics. This research was approved by the Committee of Ethics in Human Research at the Federal University of Goiás, through project submission via Brazil Platform, and was approved under number 1.643.562. All the volunteers were aware of the risks and signed the consent form.

Data accessibility. Data available from the Dryad Digital Repository: http://dx.doi.org/10.5061/dryad.443m08h [59].

Authors' contributions. L.R.B.S. collected data, wrote, reviewed and edited the manuscript. P.R.V.G. contributed with data analysis, wrote, reviewed and edited the manuscript. T.B. wrote, reviewed and edited the manuscript. M.A.B.F. collected data and contributed with the discussion. F.M.A. and M.S.S. collected data and contributed with data analysis and discussion. G.R.P. collected data and reviewed the manuscript. R.R.-C. contributed with data analysis. V.C. contributed with data analysis and reviewed the manuscript. A.C.S.R. contributed with data analysis and reviewed the manuscript.

Competing interests. We declare that there is no conflict of interest.

Funding. The authors are grateful for the financial support by agencies CAPES (Coordenação de Aperfeiçoamento de Pessoal de Nível Superior) and CNPq (Conselho Nacional de Desenvolvimento Científico e Tecnológico e Coordenação de Aperfeiçoamento de Pessoal de Nível Superior).

Acknowledgements. We would like to thank the participants for their effort and thank the funding agencies for financial support for the development of this study. P.R.V.G. receives a Research Grant from CNPq (304435/2018-0) and the project received financial support from CNPq by the grant no. 405711/2016-6.

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
