## [Reviewer comments · Royal Society Open Science]

Review History

RSOS-190639.R0 (Original submission)

Review form: Reviewer 1

Is the manuscript scientifically sound in its present form?

Yes

Are the interpretations and conclusions justified by the results?

Yes

Is the language acceptable?

Yes

Is it clear how to access all supporting data?

Yes

Reports © 2019 The Reviewers; Decision Letters © 2019 The Reviewers and Editors; Responses © 2019 The Reviewers, Editors and Authors. Published by the Royal Society under the terms of the Creative Commons Attribution License <http://creativecommons.org/licenses/by/4.0/>, which permits unrestricted use, provided the original author and source are credited

Do you have any ethical concerns with this paper?

No

Have you any concerns about statistical analyses in this paper?

No

Recommendation?

Major revision is needed (please make suggestions in comments)

Comments to the Author(s)

In this paper, the authors compare the heart rate dynamics and variability before and after moderate-intensity and high-intensity training protocols. The authors suggested that high-intensity training with short duration submaximal stimuli and moderate-intensity training might promote cardiac protection, and can be effective strategy for decreasing cardiovascular overload during exercise while working at high mechanical intensities. The paper is very well written. The methodology and results have been well-explained. Few clarifications are required:

Methods:

- Page 4, Line 99-102: The authors first reported that the sample consisted of 12 males but later mentioned that 7 were excluded. For better understanding, it is highly recommended to state how many participants did the authors approach and consented, then how many were excluded based on exclusion criteria and finally how many were included in the study that were able to perform the full protocol.
- Page 5, Line 141: High-intensity exercise can induce noise and movement artifacts into the signals being recorded. For the HRV analysis, were there any additional steps taken to test data quality other than greatest stability in RRI time series?
- Page 6, lines 173-179: Please check equations 3 and 4. the exponential time constant (τ) is missing, i.e. misrepresented by 't'. Please correct. Also equation 3, Y as a function of time $Y(t)$ and not Y subscript t.
- Page 7, lines 194-203: What parts of the statistical analysis were performed using SPSS and what parts were done with GraphPad Software? It appears that the analysis done for figure 1 was done using GraphPad while for the tables SPSS was used. Please clarify and explain the use of 2 statistical software.

Results:

- Page 8, line 229 and Page 19, figure 1: It is better to draw lines while comparing groups and showing significance between two bar plots. The symbols in the figures show significant differences but it is not clear in comparison to which group unless the caption is read. Please clarify.
- Page 8, Line 230: What is the "tau" variable? Please explain.

Discussion:

- Page 10, Lines 300-303: Please explain in detail the possible reason for the values above 30bpm after first minute of recovery.

- Page 11, Lines 324-327: This paragraph doesn't fit here and should be moved either to the first part or end of the discussion.

- In general, it is highly recommended that the authors make the discussion more concise and easy to follow.

Review form: Reviewer 2

Is the manuscript scientifically sound in its present form?

No

Are the interpretations and conclusions justified by the results?

No

Is the language acceptable?

No

Is it clear how to access all supporting data?

Yes

Do you have any ethical concerns with this paper?

No

Have you any concerns about statistical analyses in this paper?

Yes

Recommendation?

Reject

Comments to the Author(s)

Authors investigated the HRV before and after exercise. The motivation behind the study is not clear as this area has been heavily investigated.

In fact, authors wrote:

"Our findings are in line with other authors who showed that the higher the intensity of the exercise was, measured by the HRpeak, the slower the recovery of vagal activity [47,48]" So my question is it reproducibility study? It is well-known that HRV changes over the different intensity of exercises.

- 1) can you please state the hypothesis of this work?
- 2) provide a figure to show the protocol.
- 3) can you elaborate on how HR calculated? is it using ECG or PPG signals?
- 4) Table 2 is for participants at rest state?
- 5) it is not clear in figure 1 what the statistical significance refers to. Can you put the mark in between the compared bars?

6) many references are missing

Schuit, Albertine J., et al. "Exercise training and heart rate variability in older people." *Medicine and science in sports and exercise* 31.6 (1999): 816-821.

Hottenrott, Kuno, Olaf Hoos, and Hans Dieter Esperer. "Heart rate variability and physical exercise. Current status." *Herz* 31.6 (2006): 544-552.

Hynynen, E., et al. "Effects of moderate and heavy endurance exercise on nocturnal HRV." *International journal of sports medicine* 31.06 (2010): 428-432.

Pober, David M., Barry Braun, and Patty S. Freedson. "Effects of a single bout of exercise on resting heart rate variability." *Medicine and science in sports and exercise* 36.7 (2004): 1140-1148.

Perini, Renza, et al. "Aerobic training and cardiovascular responses at rest and during exercise in older men and women." *Medicine and Science in Sports and Exercise* 34.4 (2002): 700-708.

Rosenwinkel, Eric T., et al. "Exercise and autonomic function in health and cardiovascular disease." *Cardiology clinics* 19.3 (2001): 369-387.

Tulppo, Mikko P., et al. "Quantitative beat-to-beat analysis of heart rate dynamics during exercise." *American journal of physiology-heart and circulatory physiology* 271.1 (1996): H244-H252.

Review form: Reviewer 3

Is the manuscript scientifically sound in its present form?

Yes

Are the interpretations and conclusions justified by the results?

Yes

Is the language acceptable?

Yes

Is it clear how to access all supporting data?

Not Applicable

Do you have any ethical concerns with this paper?

No

Have you any concerns about statistical analyses in this paper?

No

Recommendation?

Accept with minor revision (please list in comments)

Comments to the Author(s)

General comments:

In the manuscript entitled: "Exponential model for analysis of heart rate responses and autonomic cardiac 2 modulation during different intensities of physical exercise", the authors had as objective "to compare HR responses as a surrogate of ANS balance 95 during three different exercise training protocols (two HIIT and one MICT) in healthy 96 young male individuals". The

mainly suggest that MICT promote a faster response in relation to both HIIT protocols, which makes its prescription seemingly safe in cardiovascular terms.

The introduction is well writing, but this reviewer appreciate a better justification of the study, considering that another studies compared the influence of different HIIT protocols on ANS of healthy subjects, and compared a model of HIIT and MICT on ANS of healthy subjects too. The materials and methods are clear and well writing, I just suggest that some information be inserted to increase the quality of the study. The results and discussion session are well described.

Specific comments:

a) Abstract: the abstract is clear, simple and well written.

b) Introduction: the introduction is well writing, but the literature support information about different models and HIIT or the comparison of HIIT and MICT (Castrillón CIM et al. High-Intensity Intermittent Exercise and Autonomic Modulation: Effects of Different Volume Session. Int J Sports Medicine. 2017;38(6):468-472/ Cabral-Santos et al. Impact of High-intensity Intermittent and Moderate-intensity Continuous Exercise on Autonomic Modulation in Young Men. Int J Sports Med. 2016 Jun;37(6):431-5). I understand that the present study has some different characteristics compared to the others cited above, but I appreciate that this information be inserted in the introduction session to better justify the study realization.

c) Objective: The objective is simple and clear.

d) Materials and Methods: the text is written at a reproducibility form, but some information could clarify some aspects of this session. Were smokers and alcoholic individuals included? Were series with RR intervals errors above 5% excluded from the analysis? Was the HRV data collected at the same period of the day, excluding possible circadian influences? Were the volunteers informed to not consume stimulant substances previously to HRV data collect?

e) Results: this session are clear and well reported.

f) Discussion: the discussion is clear and important information about the study are well described. As considerations, I appreciate that the authors report information about the limitations of the study. Also, the authors reported that "The MICT protocol obtained 353 intermediate values, so its applicability could be more recommended for the 354 intermediate phases of cardiac rehabilitation or situations where high mechanical 355 overload should be avoided.", and I suggest that this information be excluded. On this study the results demonstrate the influence of different models of training in healthy subjects and in subjects with cardiovascular disease this aspect was not investigate. Considering the autonomic modulations presented in subjects with cardiovascular disease, their adaptations before, during and after the exercise can be different compared to healthy subjects, so the results of the present study could be not applicable in subjects with pathological conditions, and this cannot be imply during the discussion session.

g) Conclusion: I also appreciate that the information about the use of HIIT and MICT on cardiovascular rehabilitation be excluded. See commentaries described above.

Review form: Reviewer 4

Is the manuscript scientifically sound in its present form?

No

Are the interpretations and conclusions justified by the results?

No

Is the language acceptable?

No

Is it clear how to access all supporting data?

Yes

Do you have any ethical concerns with this paper?

No

Have you any concerns about statistical analyses in this paper?

No

Recommendation?

Major revision is needed (please make suggestions in comments)

Comments to the Author(s)

In their current work „Exponential model for analysis of heart rate responses and autonomic cardiac modulation during different intensities of physical exercise“ Silva et al. investigated three different exercise protocols (two different HIIT and one MICT) and their effect on HR response in terms of dynamics and variability. The topic chosen is of interest and the findings may potentially add to the field. While I appreciate the amount of work the authors have put into their study, there are a number of problems in this study which prevent it from publication (at least in the present form). A hypothesis is missing.

First of all, this analysis seems to involve data reported previously (Eur J Sport Sci. 2019 Jun;19(5):653-660). At least the study protocol is the same. I see that in the current analysis, data of 12 participants is supposed to be shown (Abstract, Methods), while it was 10 in the earlier study. However, only table two provides an n of the analyzed subjects. This is missing in all figures and the additional tables, which is not acceptable. It could be possible to re-analyze the data of the previous study (if this is the case). However, it will have to be clarified that some of the data is already published and at least the anthropometric and CPET data will have to be presented in the Methods rather than in the results section. Some other problems that arise from this instance may remain (see below). How many days were between the subsequent protocols? Should have been more than 24 h.

Moreover, and while I highly appreciate providing original data in the repository, I accessed the original data provided and found only data of 10 participants. This is an obvious discrepancy and needs to be explained! In addition, when recalculating some of the data, I found discrepancies in the “Linear analysis of HRoff” data (Table 3).

The authors conclude that “... in the rest-exercise transition, the MICT protocol promoted a faster HR response in relation to both HIIT protocols due to the constant load characteristic of the exercise”. Currently, I cannot follow this conclusion since data on the incline of the treadmill speed per protocol is missing. It might also be that the remaining time at the presented maximal speed was very short (due to slow incline) at least in the 30 sec HIIT. This would also have affected HRmax, which is presented as one main difference between protocols (line 241: “The main findings include a higher increase in HR (HRpeak) during HIIT-4:3.”).

The protocols in the current study were matched for total distance (line 134, which is, by the way, not presented). I am not sure how this was done. I cannot calculate the distance for the 4:3 protocol since info on recovery speed is missing. But for the 30 sec protocol it appears to be 3,55 km and for the MICT 4,0 km... Moreover, the results is a 30:30 HIIT protocol with 29! repetitions, which is rather odd. Anyways, matching the three groups for total distance seems inappropriate if the primary aim of the study was to analyze HR dynamics and variability (which I doubt since currently I believe the above mentioned previous manuscript presents the primary aim of the study). Instead, for the two HIIT protocols, matching for HRmax or maximal strain would have been appropriate. Thus, one interesting question remains unanswered: What would the data have looked like if the 100% WL group had performed the exercise until HRmax. Would the results on variability and dynamic be the same? To this end, additional experimental data is needed and the

manuscript will only be acceptable if the authors can provide additional data from an appropriately matched study group.

Other major comments:

The description of the methods is poor. I believe that the protocols were performed on the same treadmill as the initial test but cannot find this information anywhere. This is also of interest since there is no information on the speed incline that was used for the treadmill. Obviously, if you are running for only 30 sec, the incline is important and it may be necessary to adjust the incline to the other protocols (where absolute speed was lower). Currently, I cannot even tell if this would have been possible since I have no information on the device used.

Poor description is also an aspect of the CPET (also in the previous publication). Information is missing on the adjustments, mode of data acquisition (bxb, mixed?) and the parameters defining VO₂max/peak etc.

Some of the data provide in the repository appear not to be normally distributed. The authors state that this had been tested, however, there is no information on how non-normal data was analyzed.

The presentation of the different study protocols lacks accuracy. If they were matched for time, overall time should be given. Also, if during the 4:3 protocol participants had three runs at 4 min and then (line 138) a 3 min cool down, how can the recovery be 3 x 3 minutes (table 1)? Was the cool down applied on top of this?

Funding : What is CAPES and CNPq?

Minor:

Language needs to be improved. For example "HRpost presented lower values in the HIIT-30:30..." Should read "participants performing...presented..."

Table three presents cardiometabolic parameters?

Has BP measurement been performed before and after the different protocols? Or is the data in table three presenting BP measures from the initial testing?

References need to be improved. Only cite own articles were appropriate.

Supporting data should have clear headings in English! and legends if abbreviations are used.

Decision letter (RSOS-190639.R0)

18-Jun-2019

Dear Miss Rebelo,

The editors assigned to your paper ("Exponential model for analysis of heart rate responses and autonomic cardiac modulation during different intensities of physical exercise") have now received comments from reviewers. We would like you to revise your paper in accordance with the referee and Associate Editor suggestions which can be found below (not including confidential reports to the Editor). Please note this decision does not guarantee eventual acceptance.

Please submit a copy of your revised paper before 11-Jul-2019. Please note that the revision deadline will expire at 00.00am on this date. If we do not hear from you within this time then it will be assumed that the paper has been withdrawn. In exceptional circumstances, extensions may be possible if agreed with the Editorial Office in advance. We do not allow multiple rounds of revision so we urge you to make every effort to fully address all of the comments at this stage. If deemed necessary by the Editors, your manuscript will be sent back to one or more of the

original reviewers for assessment. If the original reviewers are not available, we may invite new reviewers.

- Data accessibility

If you wish to submit your supporting data or code to Dryad (<http://datadryad.org/>), or modify your current submission to dryad, please use the following link:
<http://datadryad.org/submit?journalID=RSOS&manu=RSOS-190639>

- Competing interests

- Authors' contributions

- Acknowledgements

- Funding statement

on behalf of Dr Derek Abbott (Associate Editor) and R. Kerry Rowe (Subject Editor)
openscience@royalsociety.org

Comments to Author:

Reviewers' Comments to Author:

Reviewer: 1

Comments to the Author(s)

In this paper, the authors compare the heart rate dynamics and variability before and after moderate-intensity and high-intensity training protocols. The authors suggested that high-intensity training with short duration submaximal stimuli and moderate-intensity training might promote cardiac protection, and can be effective strategy for decreasing cardiovascular overload during exercise while working at high mechanical intensities. The paper is very well written. The methodology and results have been well-explained. Few clarifications are required:

Methods:

- Page 4, Line 99-102: The authors fist reported that the sample consisted of 12 males but later mentioned that 7 were excluded. For better understanding, it is highly recommended to state how many participants did the authors approach and consented, then how many were excluded based on exclusion criteria and finally how many were included in the study that were able to perform the full protocol.

- Page 5, Line 141: High-intensity exercise can induce noise and movement artifacts into the

signals being recorded. For the HRV analysis, were there any additional steps taken to test data quality other than greatest stability in RRI time series?

- Page 6, lines 173-179: Please check equations 3 and 4. the exponential time constant (τ) is missing, i.e. misrepresented by 't'. Please correct. Also equation 3, Y as a function of time $Y(t)$ and not Y subscript t.

- Page 7, lines 194-203: What parts of the statistical analysis were performed using SPSS and what parts were done with GraphPad Software? It appears that the analysis done for figure 1 was done using GraphPad while for the tables SPSS was used. Please clarify and explain the use of 2 statistical software.

Results:

- Page 8, line 229 and Page 19, figure 1: It is better to draw lines while comparing groups and showing significance between two bar plots. The symbols in the figures show significant differences but it is not clear in comparison to which group unless the caption is read. Please clarify.

- Page 8, Line 230: What is the " τ " variable? Please explain.

Discussion:

- Page 10, Lines 300-303: Please explain in detail the possible reason for the values above 30bpm after first minute of recovery.

- Page 11, Lines 324-327: This paragraph doesn't fit here and should be moved either to the first part or end of the discussion.

- In general, it is highly recommended that the authors make the discussion more concise and easy to follow.

Reviewer: 2

Comments to the Author(s)

Authors investigated the HRV before and after exercise. The motivation behind the study is not clear as this area has been heavily investigated.

In fact, authors wrote:

"Our findings are in line with other authors who showed that the higher the intensity of the exercise was, measured by the HRpeak, the slower the recovery of vagal activity [47,48]" So my question is it reproducibility study? It is well-known that HRV changes over the different intensity of exercises.

- 1) can you please state the hypothesis of this work?
- 2) provide a figure to show the protocol.
- 3) can you elaborate on how HR calculated? is it using ECG or PPG signals?
- 4) Table 2 is for participants at rest state?
- 5) it is not clear in figure 1 what the statistical significance refers to. Can you put the mark in between the compared bars?

6) many references are missing

Schuit, Albertine J., et al. "Exercise training and heart rate variability in older people." *Medicine and science in sports and exercise* 31.6 (1999): 816-821.

Hottenrott, Kuno, Olaf Hoos, and Hans Dieter Esperer. "Heart rate variability and physical exercise. Current status." *Herz* 31.6 (2006): 544-552.

Hynynen, E., et al. "Effects of moderate and heavy endurance exercise on nocturnal HRV." *International journal of sports medicine* 31.06 (2010): 428-432.

Pober, David M., Barry Braun, and Patty S. Freedson. "Effects of a single bout of exercise on resting heart rate variability." *Medicine and science in sports and exercise* 36.7 (2004): 1140-1148.

Perini, Renza, et al. "Aerobic training and cardiovascular responses at rest and during exercise in older men and women." *Medicine and Science in Sports and Exercise* 34.4 (2002): 700-708.

Rosenwinkel, Eric T., et al. "Exercise and autonomic function in health and cardiovascular disease." *Cardiology clinics* 19.3 (2001): 369-387.

Tulppo, Mikko P., et al. "Quantitative beat-to-beat analysis of heart rate dynamics during exercise." *American journal of physiology-heart and circulatory physiology* 271.1 (1996): H244-H252.

Reviewer: 3

Comments to the Author(s)

General comments:

In the manuscript entitled: "Exponential model for analysis of heart rate responses and autonomic cardiac 2 modulation during different intensities of physical exercise", the authors had as objective "to compare HR responses as a surrogate of ANS balance 95 during three different exercise training protocols (two HIIT and one MICT) in healthy 96 young male individuals". The mainly suggest that MICT promote a faster response in relation to both HIIT protocols, which makes its prescription seemingly safe in cardiovascular terms.

The introduction is well writing, but this reviewer appreciate a better justification of the study, considering that another studies compared the influence of different HIIT protocols on ANS of healthy subjects, and compared a model of HIIT and MICT on ANS of healthy subjects too. The materials and methods are clear and well writing, I just suggest that some information be inserted to increase the quality of the study. The results and discussion session are well described.

Specific comments:

a) Abstract: the abstract is clear, simple and well written.

b) Introduction: the introduction is well writing, but the literature support information about different models and HIIT or the comparison of HIIT and MICT (Castrillón CIM et al. High-Intensity Intermittent Exercise and Autonomic Modulation: Effects of Different Volume Session. *Int J Sports Medicine*. 2017;38(6):468-472/ Cabral-Santos et al. Impact of High-intensity Intermittent and Moderate-intensity Continuous Exercise on Autonomic Modulation in Young Men. *Int J Sports Med*. 2016 Jun;37(6):431-5). I understand that the present study has some different characteristics compared to the others cited above, but I appreciate that this information be inserted in the introduction session to better justify the study realization.

c) Objective: The objective is simple and clear.

d) Materials and Methods: the text is written at a reproducibility form, but some information could clarify some aspects of this session. Were smokers and alcoholic individuals included? Were series with RR intervals errors above 5% excluded from the analysis? Was the HRV data collected at the same period of the day, excluding possible circadian influences? Were the volunteers informed to not consume stimulant substances previously to HRV data collect?

e) Results: this session are clear and well reported.

f) Discussion: the discussion is clear and important information about the study are well described. As considerations, I appreciate that the authors report information about the limitations of the study. Also, the authors reported that "The MICT protocol obtained 353 intermediate values, so its applicability could be more recommended for the 354 intermediate phases of cardiac rehabilitation or situations where high mechanical 355 overload should be avoided.", and I suggest that this information be excluded. On this study the results demonstrate the influence of different models of training in healthy subjects and in subjects with cardiovascular disease this aspect was not investigate. Considering the autonomic modulations presented in subjects with cardiovascular disease, their adaptations before, during and after the exercise can be different compared to healthy subjects, so the results of the present study could be not applicable in subjects with pathological conditions, and this cannot be imply during the discussion session.

g) Conclusion: I also appreciate that the information about the use of HIIT and MICT on cardiovascular rehabilitation be excluded. See commentaries described above.

Reviewer: 4

Comments to the Author(s)

In their current work „Exponential model for analysis of heart rate responses and autonomic cardiac modulation during different intensities of physical exercise“ Silva et al. investigated three different exercise protocols (two different HIIT and one MICT) and their effect on HR response in terms of dynamics and variability. The topic chosen is of interest and the findings may potentially add to the filed. While I appreciate the amount of work the authors have put into their study, there are a number of problems in this study which prevent it from publication (at least in the present form). A hypothesis is missing.

First of all, this analysis seems to involve data reported previously (Eur J Sport Sci. 2019 Jun;19(5):653-660). At least the study protocol is the same. I see that in the current analysis, data of 12 participants is supposed to be shown (Abstract, Methods), while it was 10 in the earlier study. However, only table two provides an n of the analyzed subjects. This is missing in all figures and the additional tables, which is not acceptable. It could be possible to re-analyze the data of the previous study (if this is the case). However, it will have to be clarified that some of the data is already published and at least the anthropometric and CPET data will have to be presented in the Methods rather than in the results section. Some other problems that arise from this instance may remain (see below). How many days were between the subsequent protocols? Should have been more than 24 h.

Moreover, and while I highly appreciate providing original data in the repository, I accessed the original data provided and found only data of 10 participants. This is an obvious discrepancy and needs to be explained! In addition, when recalculating some of the data, I found discrepancies in the "Linear analysis of HRoff" data (Table 3).

The authors conclude that "... in the rest-exercise transition, the MICT protocol promoted a faster HR response in relation to both HIIT protocols due to the constant load characteristic of the exercise". Currently, I cannot follow this conclusion since data on the incline of the treadmill speed per protocol is missing. It might also be that the remaining time at the presented maximal speed was very short (due to slow incline) at least in the 30 sec HIIT. This would also have

effected HRmax, which is presented as one main difference between protocols (line 241: “The main findings include a higher increase in HR (HRpeak) during HIIT-4:3.”). The protocols in the current study were matched for total distance (line 134, which is, by the way, not presented). I am not sure how this was done. I cannot calculate the distance for the 4:3 protocol since info on recovery speed is missing. But for the 30 sec protocol it appears to be 3,55 km and for the MICT 4,0 km... Moreover, the results is a 30:30 HIIT protocol with 29! repetitions, which is rather odd. Anyways, matching the tree groups for total distance seems inappropriate if the primary aim of the study was to analyze HR dynamics and variability (which I doubt since currently I believe the above mentioned previous manuscript presents the primary aim of the study). Instead, for the two HIIT protocols, matching for HRmax or maximal strain would have been appropriate. Thus, one interesting question remains unanswered: What would the data have looked like if the 100% WL group had performed the exercise until HRmax. Would the results on variability and dynamic be the same? To this end, additional experimental data is needed and the manuscript will only be acceptable if the authors can provide additional data from an appropriately matched study group.

Other major comments:

The description of the methods is poor. I believe that the protocols were performed on the same treadmill as the initial test but cannot find this information anywhere. This is also of interest since there is no information on the speed incline that was used for the treadmill. Obviously, if you are running for only 30 sec, the incline is important and it may be necessary to adjust the incline to the other protocols (where absolute speed was lower). Currently, I cannot even tell if this would have been possible since I have no information on the device used.

Poor description is also an aspect of the CPET (also in the previous publication). Information is missing on the adjustments, mode of data acquisition (bxb, mixed?) and the parameters defining VO₂max/peak etc.

Some of the data provide in the repository appear not to be normally distributed. The authors state that this had been tested, however, there is no information on how non-normal data was analyzed.

The presentation of the different study protocols lacks accuracy. If they were matched for time, overall time should be given. Also, if during the 4:3 protocol participants had three runs at 4 min and then (line 138) a 3 min cool down, how can the recovery be 3 x 3 minutes (table 1)? Was the cool down applied on top of this?

Funding : What is CAPES and CNPq?

Minor:

Language needs to be improved. For example “HRpost presented lower values in the HIIT-30:30...” Should read “participants performing...presented...”.

Table three presents cardiometabolic parameters?

Has BP measurement been performed before and after the different protocols? Or is the data in table three presenting BP measures from the initial testing?

References need to be improved. Only cite own articles were appropriate.

Supporting data should have clear headings in English and legends if abbreviations are used.

Comments to the Authors from the Editorial Office:

For information about language editing services endorsed by the Royal Society, please follow the link below:

<https://royalsociety.org/journals/authors/language-polishing/>

Author's Response to Decision Letter for (RSOS-190639.R0)

See Appendix A.

RSOS-190639.R1 (Revision)

Review form: Reviewer 1

Is the manuscript scientifically sound in its present form?

Yes

Are the interpretations and conclusions justified by the results?

Yes

Is the language acceptable?

Yes

Do you have any ethical concerns with this paper?

No

Have you any concerns about statistical analyses in this paper?

No

Recommendation?

Accept with minor revision (please list in comments)

Comments to the Author(s)

The authors have clarified most of my concerns. A few minor comments are as follows:

Page 14, Lines 106-110: It is not clear what the author means by "the ten volunteers analysed were different because the previous studies excluded the participants...". Did the previous study also included 10 participants from the 19 but were different from the current 10? If so, it needs to be clarified and the words should be rephrased.

Page 15, Lines 126-128: There is a typo ".. were performed in the same 'an' electric .. ". Please revise the manuscript for typos before resubmitting.

Page 16, Lines 158-159: Typo: ".. were also 'instruceted avoid' copious ..."

Page 16, :Lines 184-188: The authors seem to have identified 2 types of artifacts in detection of R-peaks. How did the authors choose 30% based on the previous study? The study didn't have any exclusion criteria for participants with cardiac arrhythmia such as premature atrial/ventricular contraction which can cause a difference of >30% from previous RRi. Secondly, after identifying the artifacts, what steps were taken? Did the authors remove the artifacts from further analysis? What percentage of data were removed? Please clarify.

Page 18, Line 226. It is enough to say that "the software was also used for generating high quality/resolution graphics"

General comment: It is highly recommended that the authors revise the manuscript for typos and grammatical errors. Although the authors mentioned that "the manuscript was reviewed by the American Journal of Experts", it appears that some typos and grammatical errors still exist.

Review form: Reviewer 3

Is the manuscript scientifically sound in its present form?

Yes

Are the interpretations and conclusions justified by the results?

Yes

Is the language acceptable?

Yes

Do you have any ethical concerns with this paper?

No

Have you any concerns about statistical analyses in this paper?

No

Recommendation?

Accept as is

Comments to the Author(s)

The paper benefitted much from the changes that the authors have made as a response to the comments provided by the reviewers and will provide important contribution to the literature.

Review form: Reviewer 4

Is the manuscript scientifically sound in its present form?

No

Are the interpretations and conclusions justified by the results?

Yes

Is the language acceptable?

No

Do you have any ethical concerns with this paper?

No

Have you any concerns about statistical analyses in this paper?

No

Recommendation?

Major revision is needed (please make suggestions in comments)

Comments to the Author(s)

The authors have put some effort into the revision of the manuscript, which I appreciate. However, some critical points remain to be corrected before publication might be possible. Of note, some of my earlier comments have not or not fully been addressed.

Major:

- The hypothesis is weak and needs specification.
- The description of the study eligibility process is still weak. Pls. be exact. What were inclusion parameters, what were exclusion parameters? The delamination to the previous study is also confusing. What is meant by "...cardiorespiratory data and the present excluded due to loss of heart rate variability data." Seems to me that some words are missing? Pls. rephrase the entire paragraph.
- In my first review (original comment three) I notified the authors about mistakes in their calculations. I recalculated the data presented in the revised manuscript using the original data in the repository and there is still at least one mistake!! I demand from the authors to recheck their entire results presented here and only re-upload the data after they are 100% sure no mistakes remain. Pls. be sure to mark all corrections made in the text and tables (as not been done for corrected numbers in the table).
- In my previous comment I asked about the incline of the treadmill to understand the time participants were at maximal speed. This has not been provided, authors only deleted the sentence I quoted. This, however, is no solution to the original problem. What was the incline of the treadmill? How long were participants at maximal speed per protocol? And how did this effect the HRon/ HRoff transition per protocol? You still conclude that the HIIT-4:3 protocol induces higher HRpeak, which might only be explained by very short time at maximal speed during the 30 s protocol (Abstract, line 277). Indicate the incline of the treadmill and discuss the effect of time at maximal speed during the 30 s protocol. In the discussion pls. also reflect on the fact that not all HIIT protocols are performed on treadmills. Would results be the same if performed as 30 s SIT on track? Pls. indicate the conditions of the studies you are comparing to.
- You now present the distances per protocol in the answer to my question. Pls. also include them in table 1 as this is important (matching parameter).
- You state that you inserted the details for CPET in the methods. However, I still see no description of the adjustments, mode of data acquisition (bxb, mixed?) and the parameters defining VO2max/peak etc. This will have to be included as you use this data in the manuscript and you state that we are looking at different individuals than the previous publication.
- Concerning language/ scientific writing the authors state that: "The changes were made as requested and the manuscript was reviewed by the American Journal of Experts." If this was actually the case, they did a poor job. Pls. make sure to revise the ENTIRE manuscript to make sure language, grammar etc. is to the point. Some examples "The CPET and all physical exercise protocols were performed in the same an electric treadmill", "study. Inclusion criteria involved been physically...", "...cardiorespiratory data and the present excluded due to loss of heart rate variability data.", "Moreover, it is important to consider that the present design allow us only to bring observations that...", "As for relevance and applicability these results suggest that HIIT-4:3 protocol...".
- Supporting data still does not have clear headings in English. Legends for abbreviations are still missing.
- The study seems to be adequately powered for the delta 30 analysis but underpowered for the RMSSD analysis. Pls. comment.
- References #14 and #30 appear to be the same. Authors still try to put own references in wherever possible. This should be avoided.

Review form: Reviewer 5

Is the manuscript scientifically sound in its present form?

Yes

Are the interpretations and conclusions justified by the results?

Yes

Is the language acceptable?

Yes

Do you have any ethical concerns with this paper?

No

Have you any concerns about statistical analyses in this paper?

No

Recommendation?

Accept with minor revision (please list in comments)

Comments to the Author(s)

The authors should be applauded for the work that they did in revising this manuscript. The changes are sound, and have significantly improved it. That being said, I have a few minor nuances that I think will further strengthen this manuscript.

-Lines 85-91. I think here is the place to abbreviate the 30:30 and 3:4. They show up on the next page (line 20), but were never actually defined. Also keep in mind that order in which they are presented in this particular section are in contrast to the abstract, and how the they are presented throughout the rest of the manuscript.

-Line 102, 'been', I think you mean 'being'.

-Line 182. This sentence needs a reference. I say this, as not all data demonstrate that LF is purely sympathetic (Parati et al., 2003, for instance).

-Line 234. Please double check journal requirements, as I think that city, state are necessary for SPSS.

Review form: Reviewer 6

Is the manuscript scientifically sound in its present form?

Yes

Are the interpretations and conclusions justified by the results?

Yes

Is the language acceptable?

Yes

Do you have any ethical concerns with this paper?

No

Have you any concerns about statistical analyses in this paper?

No

Recommendation?

Accept as is

Comments to the Author(s)

The authors worried about answering all points asked by the previous reviewers which made the paper much better. The research question is relevant and contributes to the field that its inserted.

Decision letter (RSOS-190639.R1)

03-Sep-2019

Dear Miss Rebelo:

Manuscript ID RSOS-190639.R1 entitled "Exponential model for analysis of heart rate responses and autonomic cardiac modulation during different intensities of physical exercise" which you submitted to Royal Society Open Science, has been reviewed. The comments of the reviewer(s) are included at the bottom of this letter.

Please submit a copy of your revised paper before 26-Sep-2019. Please note that the revision deadline will expire at 00.00am on this date. If we do not hear from you within this time then it will be assumed that the paper has been withdrawn. In exceptional circumstances, extensions may be possible if agreed with the Editorial Office in advance. We do not allow multiple rounds of revision so we urge you to make every effort to fully address all of the comments at this stage. If deemed necessary by the Editors, your manuscript will be sent back to one or more of the original reviewers for assessment. If the original reviewers are not available we may invite new reviewers.

- Ethics statement

- Data accessibility

- Competing interests

- Authors' contributions

- Acknowledgements

- Funding statement

Kind regards,

Andrew Dunn

on behalf of Dr Derek Abbott (Associate Editor) and R. Kerry Rowe (Subject Editor)

Reviewer comments to Author:

Reviewer: 1

Comments to the Author(s)

The authors have clarified most of my concerns. A few minor comments are as follows:

Page 14, Lines 106-110: It is not clear what the author means by "the ten volunteers analysed were different because the previous studies excluded the participants...". Did the previous study also included 10 participants from the 19 but were different from the current 10? If so, it needs to be clarified and the words should be rephrased.

Page 15, Lines 126-128: There is a typo ".. were performed in the same 'an' electric .. ". Please revise the manuscript for typos before resubmitting.

Page 16, Lines 158-159: Typo: ".. were also 'instruceted avoid' copious ..."

Page 16, :Lines 184-188: The authors seem to have identified 2 types of artifacts in detection of R-peaks. How did the authors choose 30% based on the previous study? The study didn't have any exclusion criteria for participants with cardiac arrhythmia such as premature atrial/ventricular contraction which can cause a difference of >30% from previous RRi. Secondly, after identifying the artifacts, what steps were taken? Did the authors remove the artifacts from further analysis? What percentage of data were removed? Please clarify.

Page 18, Line 226. It is enough to say that "the software was also used for generating high quality/resolution graphics"

General comment: It is highly recommended that the authors revise the manuscript for typos and grammatical errors. Although the authors mentioned that "the manuscript was reviewed by the American Journal of Experts", it appears that some typos and grammatical errors still exist.

Reviewer: 4

Comments to the Author(s)

The authors have put some effort into the revision of the manuscript, which I appreciate. However, some critical points remain to be corrected before publication might be possible. Of note, some of my earlier comments have not or not fully been addressed.

Major:

- The hypothesis is weak and needs specification.
- The description of the study eligibility process is still weak. Pls. be exact. What were inclusion parameters, what were exclusion parameters? The delamination to the previous study is also confusing. What is meant by "...cardiorespiratory data and the present excluded due to loss of heart rate variability data." Seems to me that some words are missing? Pls. rephrase the entire paragraph.
- In my first review (original comment three) I notified the authors about mistakes in their calculations. I recalculated the data presented in the revised manuscript using the original data in the repository and there is still at least one mistake!! I demand from the authors to recheck their entire results presented here and only re-upload the data after they are 100% sure no mistakes remain. Pls. be sure to mark all corrections made in the text and tables (as not been done for corrected numbers in the table).

- In my previous comment I asked about the incline of the treadmill to understand the time participants were at maximal speed. This has not been provided, authors only deleted the sentence I quoted. This, however, is no solution to the original problem. What was the incline of the treadmill? How long were participants at maximal speed per protocol? And how did this effect the HRon/ HRoff transition per protocol? You still conclude that the HIIT-4:3 protocol induces higher HRpeak, which might only be explained by very short time at maximal speed during the 30 s protocol (Abstract, line 277). Indicate the incline of the treadmill and discuss the effect of time at maximal speed during the 30 s protocol. In the discussion pls. also reflect on the fact that not all HIIT protocols are performed on treadmills. Would results be the same if performed as 30 s SIT on track? Pls. indicate the conditions of the studies you are comparing to.
- You now present the distances per protocol in the answer to my question. Pls. also include them in table 1 as this is important (matching parameter).
- You state that you inserted the details for CPET in the methods. However, I still see no description of the adjustments, mode of data acquisition (bxb, mixed?) and the parameters defining VO2max/peak etc. This will have to be included as you use this data in the manuscript and you state that we are looking at different individuals than the previous publication.
- Concerning language/ scientific writing the authors state that: "The changes were made as requested and the manuscript was reviewed by the American Journal of Experts." If this was actually the case, they did a poor job. Pls. make sure to revise the ENTIRE manuscript to make sure language, grammar etc. is to the point. Some examples "The CPET and all physical exercise protocols were performed in the same an electric treadmill", "study. Inclusion criteria involved been physically...", "...cardiorespiratory data and the present excluded due to loss of heart rate variability data.", "Moreover, it is important to consider that the present design allow us only to bring observations that...", "As for relevance and applicability these results suggest that HIIT-4:3 protocol...".
- Supporting data still does not have clear headings in English. Legends for abbreviations are still missing.
- The study seems to be adequately powered for the delta 30 analysis but underpowered for the RMSSD analysis. Pls. comment.
- References #14 and #30 appear to be the same. Authors still try to put own references in wherever possible. This should be avoided.

Reviewer: 5

Comments to the Author(s)

The authors should be applauded for the work that they did in revising this manuscript. The changes are sound, and have significantly improved it. That being said, I have a few minor nuances that I think will further strengthen this manuscript.

-Lines 85-91. I think here is the place to abbreviate the 30:30 and 3:4. They show up on the next page (line 20), but were never actually defined. Also keep in mind that order in which they are presented in this particular section are in contrast to the abstract, and how the they are presented throughout the rest of the manuscript.

-Line 102, 'been', I think you mean 'being'.

-Line 182. This sentence needs a reference. I say this, as not all data demonstrate that LF is purely sympathetic (Parati et al., 2003, for instance).

-Line 234. Please double check journal requirements, as I think that city, state are necessary for SPSS.

Reviewer: 3

Comments to the Author(s)

The paper benefitted much from the changes that the authors have made as a response to the comments provided by the reviewers and will provide important contribution to the literature.

Reviewer: 6

Comments to the Author(s)

The authors worried about answering all points asked by the previous reviewers which made the paper much better. The research question is relevant and contributes to the field that its inserted.

Author's Response to Decision Letter for (RSOS-190639.R1)

See Appendix B.

Decision letter (RSOS-190639.R2)

24-Sep-2019

Dear Miss Rebelo,

I am pleased to inform you that your manuscript entitled "Exponential model for analysis of heart rate responses and autonomic cardiac modulation during different intensities of physical exercise" is now accepted for publication in Royal Society Open Science.

on behalf of Dr Derek Abbott (Associate Editor) and R. Kerry Rowe (Subject Editor)
openscience@royalsociety.org

Follow Royal Society Publishing on Twitter: [@RSocPublishing](https://twitter.com/RSocPublishing)
Follow Royal Society Publishing on Facebook:
<https://www.facebook.com/RoyalSocietyPublishing.FanPage/>
Read Royal Society Publishing's blog: <https://blogs.royalsociety.org/publishing/>

Appendix A

Dear editors and reviewers,

We would like to thank you for taking you time to review our article. We made all the requested changes (in red in the text) and below we provide the answers point by point.

Reviewer: 1

Comments to the Author(s)

In this paper, the authors compare the heart rate dynamics and variability before and after moderate-intensity and high-intensity training protocols. The authors suggested that high-intensity training with short duration submaximal stimuli and moderate-intensity training might promote cardiac protection, and can be effective strategy for decreasing cardiovascular overload during exercise while working at high mechanical intensities. The paper is very well written. The methodology and results have been well-explained. Few clarifications are required:

Methods:

- Page 4, Line 99-102: The authors fist reported that the sample consisted of 12 males but later mentioned that 7 were excluded. For better understanding, it is highly recommended to state how many participants did the authors approach and consented, then how many were excluded based on exclusion criteria and finally how many were included in the study that were able to perform the full protocol.

Answer: Thank you for the suggestion. We amended the text in order to clarify this point in the Methods section

- Page 5, Line 141: High-intensity exercise can induce noise and movement artifacts into the signals being recorded. For the HRV analysis, were there any additional steps taken to test data quality other than greatest stability in RRi time series?

Answer: Thanks. We inserted the information in the Methods

- Page 6, lines 173-179: Please check equations 3 and 4. the exponential time constant (τ) is missing, i.e. misrepresented by 't'. Please correct. Also equation 3, Y as a function of time $Y(t)$ and not Y subscript t.

Answer: Thanks for you comment. The equations were corrected accordingly

- Page 7, lines 194-203: What parts of the statistical analysis were performed using SPSS and what parts were done with GraphPad Software? It appears that the analysis done for figure 1 was done using GraphPad while for the tables SPSS was used. Please clarify and explain the use of 2 statistical software.

Answer: Thank you. We corrected the text in order to clarify.

Results:

- Page 8, line 229 and Page 19, figure 1: It is better to draw lines while comparing groups and showing significance between two bar plots. The symbols in the figures show significant differences but it is not clear in comparison to which group unless the caption is read. Please clarify.

Answer: Thanks for the suggestion, we made the changes in figure 1 in order to better and clarify the image visualization.

- Page 8, Line 230: What is the "tau" variable? Please explain.

Answer: Tau is a variable called a "time constant", it is used to describe the progression or decline of heart rate through an equation described in the Methods session.

Discussion:

- Page 10, Lines 300-303: Please explain in detail the possible reason for the values above 30bpm after first minute of recovery.

Answer: Thank you, we inserted the information.

- Page 11, Lines 324-327: This paragraph doesn't fit here and should be moved either to the first part or end of the discussion.

Answer: Thanks for your suggestion. We moved it to the first part of the discussion.

- In general, it is highly recommended that the authors make the discussion more concise and easy to follow.

Answer: Thank you, we reviewed the discussion and tried to make it more concise.

Reviewer: 2

Comments to the Author(s)

Authors investigated the HRV before and after exercise. The motivation behind the study is not clear as this area has been heavily investigated.

In fact, authors wrote:

"Our findings are in line with other authors who showed that the higher the intensity of the exercise was, measured by the HRpeak, the slower the recovery of vagal activity

[47,48]" So my question is it reproducibility study? It is well-known that HRV changes over the different intensity of exercises.

Answer: Good point. High-intensity interval training (HIIT) has been widely used. However, considering the wide variety in HIIT protocol designs, it is necessary to compare different HIIT protocols with MICT to provide a better physiological understanding of their effects and allow a better exercise training prescription. One big challenge when studying or prescribing HIIT is how to define "intensity". Some protocols might involve lower cardiovascular intensities (HR) though at a higher mechanical intensity (speed), as was the case with HIIT-30:30. In contrast, others, as HIIT-4:3, would involve higher cardiovascular intensity (HR) and lower mechanical intensity (velocity). Our aim was to test if these different arrangements would differently impact autonomic modulation and our results brought some interesting findings, since they showed that working at higher mechanical intensities with short duration might be safer, in terms of autonomic modulation. To the best of our knowledge, this is the first study to evaluate and compare two modalities of HIIT with MICT in HR dynamics and it might have important practical and clinical applications, as suggested in the discussion and conclusions.

1) can you please state the hypothesis of this work?

Answer: Thanks for your comment. We added the hypothesis in the introduction.

2) provide a figure to show the protocol.

Answer: Thank you, we inserted the information.

3) can you elaborate on how HR calculated? is it using ECG or PPG signals?

Answer: Thank you. We added a sentence in the methods to clarify that point.

4) Table 2 is for participants at rest state?

Answer: Thanks for your comment. The HRV data presented in table 2 are rest data, we include in the description.

5) it is not clear in figure 1 what the statistical significance refers to. Can you put the mark in between the compared bars?

Answer: Thanks for the suggestion, we made the changes in figure 1 in order to improve its understanding.

6) many references are missing

Schuit, Albertine J., et al. "Exercise training and heart rate variability in older people." *Medicine and science in sports and exercise* 31.6 (1999): 816-821.

Hottenrott, Kuno, Olaf Hoos, and Hans Dieter Esperer. "Heart rate variability and

physical exercise. Current status." Herz 31.6 (2006): 544-552.

Hynynen, E., et al. "Effects of moderate and heavy endurance exercise on nocturnal HRV." International journal of sports medicine 31.06 (2010): 428-432.

Pober, David M., Barry Braun, and Patty S. Freedson. "Effects of a single bout of exercise on resting heart rate variability." Medicine and science in sports and exercise 36.7 (2004): 1140-1148.

Perini, Renza, et al. "Aerobic training and cardiovascular responses at rest and during exercise in older men and women." Medicine and Science in Sports and Exercise 34.4 (2002): 700-708.

Rosenwinkel, Eric T., et al. "Exercise and autonomic function in health and cardiovascular disease." Cardiology clinics 19.3 (2001): 369-387.

Tulppo, Mikko P., et al. "Quantitative beat-to-beat analysis of heart rate dynamics during exercise." American journal of physiology-heart and circulatory physiology 271.1 (1996): H244-H252.

Answer: Thanks for you indication. We have inserted the references

Reviewer: 3

Comments to the Author(s)
General comments:

In the manuscript entitled: "Exponential model for analysis of heart rate responses and autonomic cardiac 2 modulation during different intensities of physical exercise", the authors had as objective "to compare HR responses as a surrogate of ANS balance 95 during three different exercise training protocols (two HIIT and one MICT) in healthy 96 young male individuals". The mainly suggest that MICT promote a faster response in relation to both HIIT protocols, which makes its prescription seemingly safe in cardiovascular terms.

The introduction is well writing, but this reviewer appreciate a better justification of the study, considering that another studies compared the influence of different HIIT protocols on ANS of healthy subjects, and compared a model of HIIT and MICT on ANS of healthy subjects too. The materials and methods are clear and well writing, I just suggest that some information be inserted to increase the quality of the study. The results and discussion session are well described.

Specific comments:

a) Abstract: the abstract is clear, simple and well written.

Answer: Thanks for your comment.

b) Introduction: the introduction is well writing, but the literature support information about different models and HIIT or the comparison of HIIT and MICT (Castrillón CIM et al. High-Intensity Intermittent Exercise and Autonomic Modulation: Effects of Different Volume Session. Int J Sports Medicine. 2017;38(6):468-472/ Cabral-Santos et al. Impact of High-intensity Intermittent and Moderate-intensity Continuous Exercise on Autonomic Modulation in Young Men. Int J Sports Med. 2016 Jun;37(6):431-5). I understand that the present study has some different characteristics compared to the others cited above, but I appreciate that this information be inserted in the introduction session to better justify the study realization.

Answer: Thaks, this information has been added to the text.

c) Objective: The objective is simple and clear.

Answer: Thanks for your comment.

d) Materials and Methods: the text is written at a reproducibility form, but some information could clarify some aspects of this session. Were smokers and alcoholic individuals included? Were series with RR intervals errors above 5% excluded from the analysis? Was the HRV data collected at the same period of the day, excluding possible circadian influences? Were the volunteers informed to not consume stimulant substances previously to HRV data collect?

Answer: Thanks for your comment. Smokers and/or alcoholic individuals were excluded, this information was added in Méthods (Participants). The exclusion of HRV artifacts and the information about the time of day for collection and previous recommendations to volunteers has also been inserted in the Methods ("Heart Rate Variability (HRV) Recording and Analysis").

e) Results: this session are clear and well reported.

Answer: Thanks for your comment.

f) Discussion: the discussion is clear and important information about the study are well described. As considerations, I appreciate that the authors report information about the limitations of the study. Also, the authors reported that “The MICT protocol obtained 353 intermediate values, so its applicability could be more recommended for the 354 intermediate phases of cardiac rehabilitation or situations where high mechanical 355 overload should be avoided.”, and I suggest that this information be excluded. On this study the results demonstrate the influence of different models of training in healthy subjects and in subjects with cardiovascular disease this aspect was not investigate. Considering the autonomic modulations presented in subjects with cardiovascular disease, their adaptations before, during and after the exercise can be different compared

to healthy subjects, so the results of the present study could be not applicable in subjects with pathological conditions, and this cannot be imply during the discussion session.

Answer: Thanks for your comment. we edited the text to in accordance with your suggestions.

g) Conclusion: I also appreciate that the information about the use of HIIT and MICT on cardiovascular rehabilitation be excluded. See commentaries described above.

Answer: Thanks for your comment. The information was removed.

Reviewer: 4

Comments to the Author(s)

In their current work „Exponential model for analysis of heart rate responses and autonomic cardiac modulation during different intensities of physical exercise“ Silva et al. investigated three different exercise protocols (two different HIIT and one MICT) and their effect on HR response in terms of dynamics and variability. The topic chosen is of interest and the findings may potentially add to the filed. While I appreciate the amount of work the authors have put into their study, there are a number of problems in this study which prevent it from publication (at least in the present form). A hypothesis is missing.

Answer: Thanks for your comment. We inserted the hypothesis in the text.

First of all, this analysis seems to involve data reported previously (Eur J Sport Sci. 2019 Jun;19(5):653-660). At least the study protocol is the same. I see that in the current analysis, data of 12 participants is supposed to be shown (Abstract, Methods), while it was 10 in the earlier study. However, only table two provides an n of the analyzed subjects. This is missing in all figures and the additional tables, which is not acceptable. It could be possible to re-analyze the data of the previous study (if this is the case). However, it will have to be clarified that some of the data is already published and at least the anthropometric and CPET data will have to be presented in the Methods rather than in the results section. Some other problems that arise from this instance may remain (see below). How many days were between the subsequent protocols? Should have been more than 24 h.

Answer: We appreciate you comments and have inserted all the requested information in the text. Importantly, we have clarified that the initial 19 volunteers were the same to a previous study, however, the ten volunteers analysed were different because the previous study excluded the participants due to loss of cardiorespiratory data and the present excluded due to loss of heart rate variability data.

Moreover, and while I highly appreciate providing original data in the repository, I accessed the original data provided and found only data of 10 participants. This is an obvious discrepancy and needs to be explained! In addition, when recalculating some of the data, I found discrepancies in the “Linear analysis of HRoff” data (Table 3).

Answer: We are really sorry about that, we reviewed the data and made the necessary corrections.

The authors conclude that “... in the rest-exercise transition, the MICT protocol promoted a faster HR response in relation to both HIIT protocols due to the constant load characteristic of the exercise”. Currently, I cannot follow this conclusion since data on the incline of the treadmill speed per protocol is missing. It might also be that the remaining time at the presented maximal speed was very short (due to slow incline) at least in the 30 sec HIIT. This would also have effected HRmax, which is presented as one main difference between protocols (line 241: “The main findings include a higher increase in HR (HRpeak) during HIIT-4:3.”).

Answer: We completely agree. We removed the sentence.

The protocols in the current study were matched for total distance (line 134, which is, by the way, not presented). I am not sure how this was done. I cannot calculate the distance for the 4:3 protocol since info on recovery speed is missing. But for the 30 sec protocol it appears to be 3,55 km and for the MICT 4,0 km... Moreover, the results is a 30:30 HIIT protocol with 29! repetitions, which is rather odd. Anyways, matching the tree groups for total distance seems inappropriate if the primary aim of the study was to analyze HR dynamics and variability (which I doubt since currently I believe the above mentioned previous manuscript presents the primary aim of the study). Instead, for the two HIIT protocols, matching for HRmax or maximal strain would have been appropriate. Thus, one interesting question remains unanswered: What would the data have looked like if the 100% WL group had performed the exercise until HRmax. Would the results on variability and dynamic be the same? To this end, additional experimental data is needed and the manuscript will only be acceptable if the authors can provide additional data from an appropriately matched study group.

Answer: equating for distance was the solution found for standardization, since the protocols were performed at constant intensity. Therefore, the variations in HR were inherent to them.

The protocols have been described inadequately in the previous version. We are really sorry, but now they are correctly presented in table 1. We shall take a participant that reached VOpeak at 15km/h as an example (excluding warm up and cool down):

MICT – 21min at 10,5km/h – would lead to 3.675km

HIIT-30:30 29 bouts of 30s (14.5min) at 15km/h – lead to 3.625km

HIIT-4:3 4 bouts of 3min (12min) at 13.5km/h + 2 intervals of 3min (6min) at 9km/h – lead to 3.6km

Other major comments:

The description of the methods is poor. I believe that the protocols were performed on the same treadmill as the initial test but cannot find this information anywhere. This is also of interest since there is no information on the speed incline that was used for the treadmill. Obviously, if you are running for only 30 sec, the incline is important and it may be necessary to adjust the incline to the other protocols (where absolute speed was lower). Currently, I cannot even tell if this would have been possible since I have no information on the device used.

Answer: Thank you. We inserted the information in the Methods section.

Poor description is also an aspect of the CPET (also in the previous publication). Information is missing on the adjustments, mode of data acquisition (bxb, mixed?) and the parameters defining VO₂max/peak etc.

Answer: Thank you. We inserted the information in the Methods section.

Some of the data provide in the repository appear not to be normally distributed. The authors state that this had been tested, however, there is no information on how non-normal data was analyzed.

Answer: Thanks for your comment, we inserted the information in the statistical analysis.

The presentation of the different study protocols lacks accuracy. If they were matched for time, overall time should be given. Also, if during the 4:3 protocol participants had three runs at 4 min and then (line 138) a 3 min cool down, how can the recovery be 3 x 3 minutes (table 1)? Was the cool down applied on top of this?

Response. Sorry for the mistake. Table 1 was corrected accordingly.

Funding : What is CAPES and CNPq?

Answer: They are Brazilian funding agencies; we have clarified the acronyms.

Minor:

Language needs to be improved. For example “HRpost presented lower values in the HIIT-30:30...” Should read “participants performing...presented...”.

Answer: We appreciate your feedback. The changes were made as requested and the manuscript was reviewed by the American Journal of Experts.

Table three presents cardiometabolic parameters?

Answer: Table 3 shows only cardiovascular data such as blood pressure and heart rate, we corrected the text.

Has BP measurement been performed before and after the different protocols? Or is the data in table three presenting BP measures from the initial testing?

Answer: Systemic blood pressure was assessed at pre, during and after each protocol.

References need to be improved. Only cite own articles were appropriate.

Answer: We appreciate your feedback and reviewed the text.

Supporting data should have clear headings in English and legends if abbreviations are used.

Answer: We appreciate your feedback. We changed the text accordingly

Appendix B

Dear editors and reviewers,

We would like to thank you for taking you time to review our article. We made all the requested changes (in red in the text) and below we provide the answers point by point.

Reviewer: 1

Comments to the Author(s)

The authors have clarified most of my concerns. A few minor comments are as follows:

Page 14, Lines 106-110: It is not clear what the author means by "the ten volunteers analysed were different because the previous studies excluded the participants...". Did the previous study also included 10 participants from the 19 but were different from the current 10? If so, it needs to be clarified and the words should be rephrased.

Reply: Thanks. This information has been changed in the text.

Page 15, Lines 126-128: There is a typo ".. were performed in the same 'an' electric .. ". Please revise the manuscript for typos before resubmitting.

Reply: The sentence has been corrected.

Page 16, Lines 158-159: Typo: ".. were also 'instruceted avoid' copious ..."

Reply: The sentence has been corrected.

Page 16, :Lines 184-188: The authors seem to have identified 2 types of artifacts in detection of R-peaks. How did the authors choose 30% based on the previous study? The study didn't have any exclusion criteria for participants with cardiac arrhythmia such as premature atrial/ventricular contraction which can cause a difference of >30% from previous RRi. Secondly, after identifying the artifacts, what steps were taken? Did the authors remove the artifacts from further analysis? What percentage of data were removed? Please clarify.

Reply: Thanks for your comment. The choice of 30% was due to the high intensity of physical exercise. Those who presented uncontrolled cardiac arrhythmia during CPET were excluded, information included in the item methods, lines 107-109. The Kubios HRV analysis software has several filters to remove these artifacts, however before all analysis in the software mentioned above, the data goes through a manual editing process in which ectopic beats, arrhythmias and / or noise were selected and replaced by R- Adjacent laugh that shows greater normality. The percentage of data removed was 18 to 22%

Page 18, Line 226. It is enough to say that "the software was also used for generating high quality/resolution graphics"

Reply: Thanks. This information has been changed in the text.

General comment: It is highly recommended that the authors revise the manuscript for typos and grammatical errors. Although the authors mentioned that "the manuscript was reviewed

by the American Journal of Experts", it appears that some typos and grammatical errors still exist.

Reply: Thanks for the suggestion, the manuscript has been sent for further revision of the language.

Reviewer: 4

Comments to the Author(s)

The authors have put some effort into the revision of the manuscript, which I appreciate. However, some critical points remain to be corrected before publication might be possible. Of note, some of my earlier comments have not or not fully been addressed.

Major:

- The hypothesis is weak and needs specification.

Reply: Thanks. The text of the hypothesis has been changed.

- The description of the study eligibility process is still weak. Pls. be exact. What were inclusion parameters, what were exclusion parameters? The delamination to the previous study is also confusing. What is meant by "...cardiorespiratory data and the present excluded due to loss of heart rate variability data." Seems to me that some words are missing? Pls. rephrase the entire paragraph.

Reply: The paragraph was rewritten.

- In my first review (original comment three) I notified the authors about mistakes in their calculations. I recalculated the data presented in the revised manuscript using the original data in the repository and there is still at least one mistake!! I demand from the authors to recheck their entire results presented here and only re-upload the data after they are 100% sure no mistakes remain. Pls. be sure to mark all corrections made in the text and tables (as not been done for corrected numbers in the table).

Reply: Thanks for your comment, all data has been revised and changed as needed, in the data where changes have been made new statistical analyzes have been made.

- In my previous comment I asked about the incline of the treadmill to understand the time participants were at maximal speed. This has not been provided, authors only deleted the sentence I quoted. This, however, is no solution to the original problem. What was the incline of the treadmill? How long were participants at maximal speed per protocol? And how did this effect the HRon/ HRoff transition per protocol? You still conclude that the HIIT-4:3 protocol induces higher HRpeak, which might only be explained by very short time at maximal speed during the 30 s protocol (Abstract, line 277). Indicate the incline of the treadmill and discuss the effect of time at maximal speed during the 30 s protocol. In the discussion pls. also reflect on

the fact that not all HIIT protocols are performed on treadmills. Would results be the same if performed as 30 s SIT on track? Pls. indicate the conditions of the studies you are comparing to.

Reply. We inserted the information that “All tests were performed without inclination to minimize the errors during the execution of the protocol and to avoid different physiological responses.” The time at each velocity is detailed in the methods, in Table 1.

We agree that the short time spent in the 30s protocol explain the low HR response, and this have been discussed (“At HIIT-30:30, despite the initial mechanical intensity being higher (100% of $WL\dot{V}O_2\text{max}$), the short duration of the effort seems to have generated less of a need for sympathetic activation, which may explain the lack of difference for HIIT-4:3.”

We included the discussion about the specificity of the treadmill.

- You now present the distances per protocol in the answer to my question. Pls. also include them in table 1 as this is important (matching parameter).

Reply: Thanks. This information has been changed in the table 1.

- You state that you inserted the details for CPET in the methods. However, I still see no description of the adjustments, mode of data acquisition (bxb, mixed?) and the parameters defining $VO_2\text{max/peak}$ etc. This will have to be included as you use this data in the manuscript and you state that we are looking at different individuals than the previous publication.

Reply: The information was entered as requested and available in the factory information (Lines 144-148).

- Concerning language/ scientific writing the authors state that: “The changes were made as requested and the manuscript was reviewed by the American Journal of Experts.” If this was actually the case, they did a poor job. Pls. make sure to revise the ENTIRE manuscript to make sure language, grammar etc. is to the point. Some examples “The CPET and all physical exercise protocols were performed in the same an electric treadmill”, “study. Inclusion criteria involved been physically...”, “...cardiorespiratory data and the present excluded due to loss of heart rate variability data.”, “Moreover, it is important to consider that the present design allow us only to bring observations that...”, “As for relevance and applicability these results suggest that HIIT-4:3 protocol...”.

Reply: Thanks for the suggestion, the manuscript has been sent for further revision of the language.

- Supporting data still does not have clear headings in English. Legends for abbreviations are still missing.

Reply: Thanks for the suggestion, the manuscript has been sent for further revision of the language.

- The study seems to be adequately powered for the delta 30 analysis but underpowered for the RMSSD analysis. Pls. comment.

Reply: In the current literature, numerous studies have shown the good reproducibility of the RMSSD index during physical exercise, and this index is sensitive to changes in intensity along

physical exertion. As an example we can cite, initially, the study by Alonso (1998) that when evaluating healthy young individuals observed reduction in RMSSD with the exercise time and increase of load.

During physical exertion innumerable changes are noticed in all body systems, promoting changes in HR. The study by Marães (2003) observed that there is an increase in HR until they can observe the loss of linearity of the ventilatory threshold, since these changes will lead to reductions in HRV values, compared to the resting state of individuals as observed in our study. To reaffirm the validity of using the RMSSD index, MacNames and Naboy (2006) stated that the RMSSD showed good agreement with RRi values in healthy adults. To ensure reliable values for the behavior of HR and HRV during physical exercise we used linear analysis through the HRV time domain, and through the analysis of HR deltas.

References:

ALONSO, D. O et al. Heart rate response and its variability during different phases of maximal graded exercise. *Arquivos Brasileiros de Cardiologia*, v. 71, n. 6, p. 787-792, 1998.

MARÃES, V. R. F. S. et al. Determinação e validação do limiar de anaerobiose a partir de métodos de análise de frequência cardíaca e de sua variabilidade. *Revista da Sociedade de Cardiologia do Estado de São Paulo*, v.13, n.4, p. 1-16, 2003.

McNames J, Aboy M. Reliability and accuracy of heart rate variability metrics versus ECG segment duration. *Med Biol Eng Comput* 2006; 44: 747–756, doi: 10.1007/s11517-006-0097-2. [Links]

Salahuddin L, Cho J, Jeong MG, Kim D. Ultra short term analysis of heart rate variability for monitoring mental stress in mobile settings. *IEEE* 2007; 23–26.

- References #14 and #30 appear to be the same. Authors still try to put own references in wherever possible. This should be avoided.

Reply: this was a problem with Mendeley, we removed the reference.

Reviewer: 5

Comments to the Author(s)

The authors should be applauded for the work that they did in revising this manuscript. The changes are sound, and have significantly improved it. That being said, I have a few minor nuances that I think will further strengthen this manuscript.

-Lines 85-91. I think here is the place to abbreviate the 30:30 and 3:4. They show up on the next page (line 20), but were never actually defined. Also keep in mind that order in which they are

presented in this particular section are in contrast to the abstract, and how they are presented throughout the rest of the manuscript.

Reply: Thanks. This information has been changed in the text (Introduction - Lines 86-91).

-Line 102, 'been', I think you mean 'being'.

Reply: Information was inserted in text (Section Methods - Line 102)

-Line 182. This sentence needs a reference. I say this, as not all data demonstrate that LF is purely sympathetic (Parati et al., 2003, for instance).

Reply: Thanks for your suggestion. The reference has been added at the end of the sentence (Line-184).

-Line 234. Please double check journal requirements, as I think that city, state are necessary for SPSS.

Reply: The request has been inserted.

Reviewer: 3

Comments to the Author(s)

The paper benefitted much from the changes that the authors have made as a response to the comments provided by the reviewers and will provide important contribution to the literature.

Reviewer: 6

Comments to the Author(s)

The authors worried about answering all points asked by the previous reviewers which made the paper much better. The research question is relevant and contributes to the field that its inserte